# TT-seq captures enhancer landscapes immediately after T-cell stimulation

Margaux Michel[1],[†] (ID), Carina Demel[1],[†] (ID), Benedikt Zacher[2], Björn Schwalb[1], Stefan Krebs[2], Helmut Blum[2], Julien Gagneur[3],* (ID) & Patrick Cramer[1],** (ID)

## Abstract

To monitor transcriptional regulation in human cells, rapid changes in enhancer and promoter activity must be captured with high sensitivity and temporal resolution. Here, we show that the recently established protocol TT-seq ("transient transcriptome sequencing") can monitor rapid changes in transcription from enhancers and promoters during the immediate response of T cells to ionomycin and phorbol 12-myristate 13-acetate (PMA). TT-seq maps eRNAs and mRNAs every 5 min after T-cell stimulation with high sensitivity and identifies many new primary response genes. TT-seq reveals that the synthesis of 1,601 eRNAs and 650 mRNAs changes significantly within only 15 min after stimulation, when standard RNA-seq does not detect differentially expressed genes. Transcription of enhancers that are primed for activation by nucleosome depletion can occur immediately and simultaneously with transcription of target gene promoters. Our results indicate that enhancer transcription is a good proxy for enhancer regulatory activity in target gene activation, and establish TT-seq as a tool for monitoring the dynamics of enhancer landscapes and transcription programs during cellular responses and differentiation.

**Keywords** enhancers; functional genomics; promoters; T-cell response; transcriptome analysis

**Subject Categories** Chromatin, Epigenetics, Genomics & Functional Genomics; Genome-Scale & Integrative Biology; Transcription

**Mol Syst Biol. (2017) 13: 920**

## Introduction

In metazoan cells, the synthesis of mRNAs from protein-coding genes during transcription is driven from promoters and activated by enhancers (Lenhard *et al*, 2012; Levine *et al*, 2014). Enhancers are regulatory units in the genome that contain binding sites for sequence-specific transcription factors and can activate mRNA transcription over long distances (Banerji *et al*, 1981). Active enhancers adopt an open chromatin structure (Calo & Wysocka, 2013) and recruit co-activators such as Mediator (Fan *et al*, 2006). Mediator can apparently bridge between enhancers and promoters because it binds both transcriptional activators and the RNA polymerase II (Pol II) initiation complex at the promoter (Malik & Roeder, 2010; Liu *et al*, 2013). Promoter–enhancer interaction ("pairing") increases initiation complex stability and promotes Pol II escape from the promoter (Splinter *et al*, 2006; DeMare *et al*, 2013; Allen & Taatjes, 2015). Promoter–enhancer pairing requires DNA looping that is facilitated within insulated neighborhoods, which are genomic regions formed by looping of DNA between two CTCF-binding sites co-occupied by cohesin (Phillips-Cremins *et al*, 2013; Dowen *et al*, 2014; Hnisz *et al*, 2016a).

The genomewide identification of enhancers is crucial for studying cellular regulation and differentiation, but remains technically challenging (Shlyueva *et al*, 2014). Enhancers may be distinguished from other genomic regions through a signature of histone modifications that can be mapped by chromatin immunoprecipitation (ChIP; Heintzman *et al*, 2007; Schübeler, 2007; Visel *et al*, 2009) or DNA accessibility assays (Xi *et al*, 2007; Thurman *et al*, 2012; Shlyueva *et al*, 2014). Regulatory active enhancers may be identified through their transcriptional activity, which is thought to be a good proxy for their function in promoter activation (Melgar *et al*, 2011; Wu *et al*, 2014; Li *et al*, 2016). Transcribed enhancers produce enhancer RNAs (eRNAs) (Kim *et al*, 2010; Djebali *et al*, 2012), which are difficult to detect because they are short-lived (Rabani *et al*, 2014; Schwalb *et al*, 2016), rapidly degraded by the exosome (Lubas *et al*, 2015), and generally not conserved over species (Andersson *et al*, 2014).

The role of enhancer transcription and/or eRNAs remains unclear (Li *et al*, 2016). It is likely that the process of enhancer transcription has a functional role, maybe in recruiting chromatin remodelers through their association with transcribing Pol II (Gribnau *et al*, 2000). Consistent with this model, enhancer transcription can precede target gene transcription (De Santa *et al*, 2010; Kaikkonen *et al*, 2013; Schaukowitch *et al*, 2014; Arner *et al*, 2015). It is also possible that eRNAs themselves have a function, because

1   Department of Molecular Biology, Max Planck Institute for Biophysical Chemistry, Göttingen, Germany
2   Gene Center Munich, Ludwig-Maximilians-Universität München, Munich, Germany
3   Department of Informatics, Technische Universität München, Garching, Germany
    *Corresponding author. Tel: +49 89 289 19411; E-mail: gagneur@in.tum.de
    **Corresponding author. Tel: +49 551 201 2800; E-mail: patrick.cramer@mpibpc.mpg.de
    †These authors contributed equally to this work

eRNA knockdown may impair target gene activation (Li *et al*, 2013; Ilott *et al*, 2014; Schaukowitch *et al*, 2014). eRNA knockdown may also have negative effects on promoter–enhancer pairing (Li *et al*, 2013), although some studies came to different conclusions (Hah *et al*, 2013; Schaukowitch *et al*, 2014).

Here, we investigate the relationship between transcription from enhancers and promoters during the human T-cell response. Upon T-cell stimulation, the T-cell receptor and the costimulatory receptor CD28 are activated, leading to a signaling cascade (Smith-Garvin *et al*, 2009). Phosphorylation of multiple factors at the plasma membrane leads to recruitment and activation of PLC-γ that cleaves $PI(4,5)P_2$ in DAG and $IP_3$. First, DAG binds and activates PKCθ, which leads to activation and nuclear translocation of the transcription factors NF-κB and AP-1. Second, $IP_3$ diffuses away from the plasma membrane and activates calcium channel receptors on the ER, increasing intracellular calcium ion concentration and leading to activation of calmodulin and calcineurin. Calcineurin activates NFAT that translocates to the nucleus and drives gene activation. T-cell stimulation via the T-cell receptor and CD28 can be mimicked by addition of PMA and ionomycin because phorbol esters activate PKC and calcium ionophores raise intracellular calcium levels (Weiss & Imboden, 1987).

The T-cell response involves rapid changes in gene expression (Marrack *et al*, 2000; Rogge *et al*, 2000; Feske *et al*, 2001; Diehn *et al*, 2002; Raghavan *et al*, 2002; Cheadle *et al*, 2005). Responding genes were classified into immediate-early, early, and late response genes based on changes in RNA levels. Immediate-early response genes are transiently activated within the first hour after stimulation (Bahrami & Drablos, 2016). There are ~40 immediate-early genes described, most of which code for transcription factors such as *FOS, FOSB, FRA1, JUNB, JUN, NFAT, NFKB,* and *EGR1* (Greenberg & Ziff, 1984; Sheng & Greenberg, 1990). Several hours after stimulation, immediate-early factors activate early and late response genes, including cytokines such as IL-2, TGF-β, or IFN-γ (Crabtree, 1989; Ellisen *et al*, 2001). Despite these studies, the immediate T-cell response and the primary events after T-cell stimulation remain incompletely understood.

To monitor immediate transcriptional changes after T-cell stimulation, we use here transient transcriptome sequencing (TT-seq). TT-seq was developed recently to detect short-lived RNAs such as eRNAs in human cells and to estimate RNA synthesis and degradation rates (Schwalb *et al*, 2016). TT-seq involves short, 5-min labeling of nascent RNA with 4-thiouridine (4sU). RNA is then fragmented, and the labeled RNA fragments are sequenced, providing a genomewide view of RNA synthesis during the 5-min labeling pulse. TT-seq is a sensitive method to detect eRNAs, because it has higher sensitivity than RNA-seq in detecting short-lived RNAs and because it is more sensitive than standard 4sU labeling in detecting short RNAs because its fragmentation step confers a transcript length-independent sampling of the nascent transcriptome (Schwalb *et al*, 2016). Hence, TT-seq should be ideally suited to map changes in eRNA and mRNA production during transcriptional activation, but this was not yet demonstrated.

Here, we use TT-seq to monitor the immediate T-cell response over the first 15 min after cell stimulation. We identify new immediate, direct target genes of the T-cell response and show that activation of immediate enhancers and promoters, as defined by RNA production, occurs simultaneously. The results also establish TT-seq as a simple-to-use, very sensitive tool to investigate transcriptional responses at high temporal resolution, ideally suited to monitor rapid changes in enhancer landscapes and in transcriptional programs during cellular differentiation and reprogramming.

# Results

## Monitoring the immediate T-cell response

We monitored immediate changes in RNA synthesis in Jurkat T cells during the first 15 min after stimulation with ionomycin and PMA using both TT-seq and RNA-seq (Fig 1A, Materials and Methods, Appendix Figs S1 and S2). We selected time points before stimulation (0 min), and 5, 10, and 15 min after stimulation. The TT-seq data revealed strong up- and downregulation of mRNA synthesis for immediately responding genes (Fig 1B and C). In TT-seq data, we also observed a high coverage of intronic regions and regions downstream of the poly-adenylation site (PAS, annotated by GENCODE; Harrow *et al*, 2012), demonstrating that TT-seq could trap short-lived RNA (Fig 1B and C).

We combined the TT-seq data to segment the genome into transcribed and non-transcribed regions using GenoSTAN (Zacher *et al*, 2017). Then, we automatically annotated a total of 22,141 transcribed regions ("transcripts") before and after T-cell stimulation (RPK cutoff = 16.5, Materials and Methods, Fig EV1 and Table EV1). Comparison with the GENCODE annotation (Harrow *et al*, 2012) enabled us to classify our annotated transcripts into 8,878 mRNAs, 590-long non-coding RNAs (lincRNAs), by requiring at least 20% of the transcribed region to overlap with GENCODE-annotated "protein_coding" or "lincRNA" (long, intervening non-coding RNA that can be found in evolutionarily conserved, intergenic regions) transcripts (Materials and Methods). The 12,673 remaining transcripts we categorized as non-coding RNAs (ncRNAs) (Fig EV1C). These RNAs may contain additional long non-coding RNAs that do not fall into GENCODE's "lincRNA" definition (Materials and Methods). The length distribution of RNAs in these classes (Fig EV1D) resembled that in our previous study of human K562 cells (Schwalb *et al*, 2016). Steady-state RNA synthesis rate and half-life distributions also agreed with previous results (Fig EV2). Taken together, we obtained a transcriptome annotation for T cells that included both stable transcripts present during steady-state growth and short-lived RNAs that are produced immediately after stimulation.

## TT-seq uncovers many immediate response genes

We next analyzed changes in transcript coverage upon T-cell stimulation after integrating reads over transcribed units at different time points (Table EV2). When we compared time points 5, 10, and 15 min with time point 0 min, RNA-seq data did not reveal any significant (FC > 2, adjusted *P*-value < 0.05, Materials and Methods) changes. In contrast, TT-seq uncovered hundreds of newly synthesized transcripts with significantly changed signals already after 5 min, and thousands of changed transcripts after 15 min following stimulation (FC > 2, adjusted *P*-value < 0.05, Materials and Methods, Fig 2A and Appendix Tables S1 and S2). These results show that transcription activity in T cells changes immediately upon

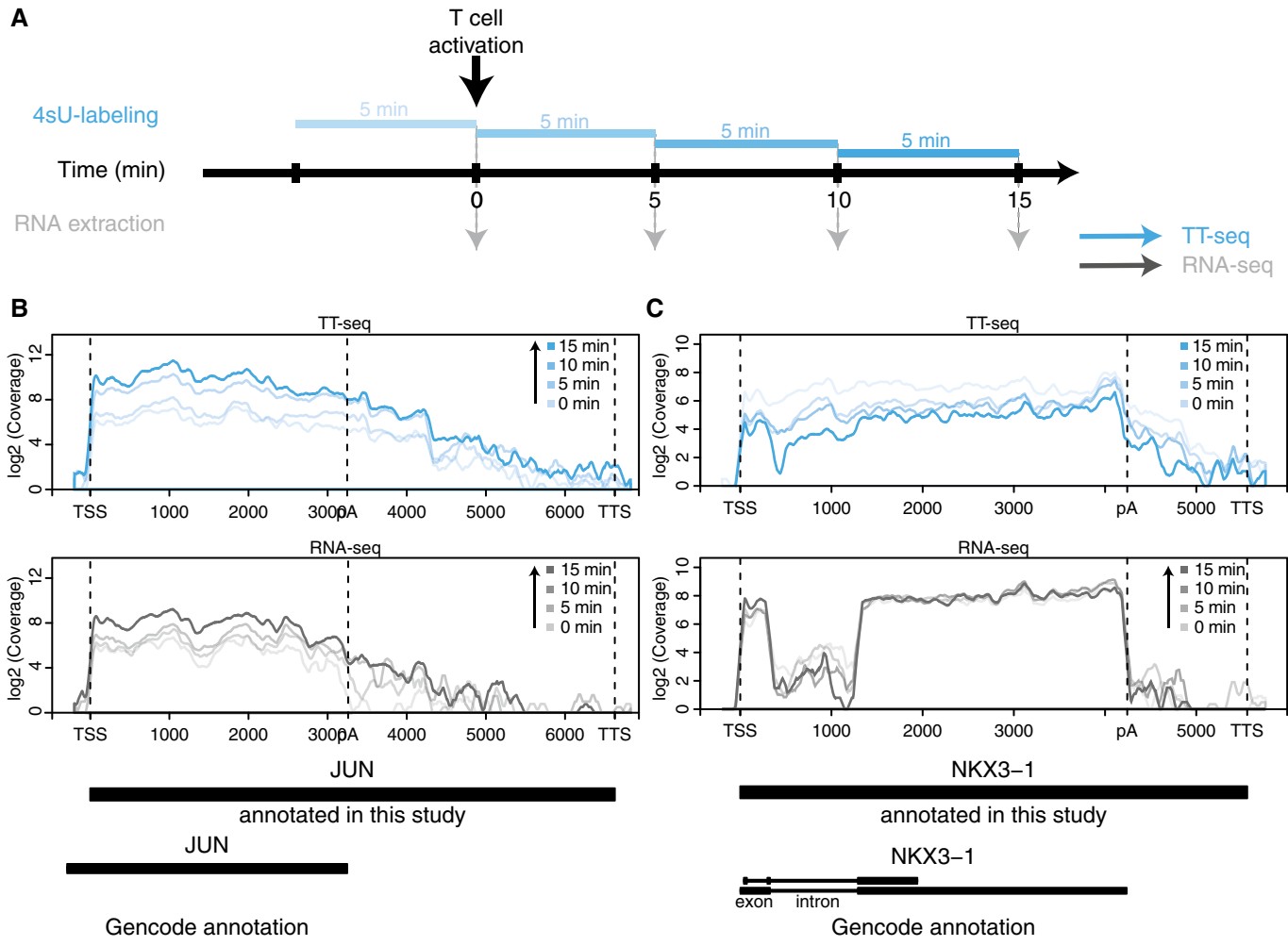

**Figure 1. TT-seq analysis of immediate response to T-cell stimulation.**

A Experimental design. RNA in cells was labeled with 4-thiouridine (4sU) for consecutive 5-min intervals. Total and 4sU-labeled RNA was extracted before T-cell stimulation and 5, 10, and 15 min after T-cell stimulation and subjected to deep-sequencing.

B, C Exemplary genome browser views for (B) an upregulated mRNA (*JUN*) and (C) a downregulated mRNA (*NKX3-1*). Blue coverage: TT-seq data for 0, 5, 10, and 15 min after stimulation; gray coverage: total RNA-seq data for 0, 5, 10 and 15 min after stimulation. TSS: transcription start site, TTS: transcription termination site, pA: poly-adenylation site.

stimulation and that TT-seq captures transcriptional up- and down-regulation with great sensitivity long before changes in RNA levels are detected by RNA-seq.

Out of a total of 3,744 transcripts that showed significantly changed synthesis 15 min after stimulation, 638 were mRNAs, and 2,986 were ncRNAs, including 120 lincRNAs (Appendix Tables S1 and S2). Many upregulated mRNAs encode known marker proteins of T-cell activation such as *FOS, FOSB, JUN, JUNB* and *CD69* (Table EV3). Other upregulated mRNAs stemmed from known immediate-early response genes, such as transcription factors *EGR1, EGR2, EGR3,* and *NR4A1,* and the stem cell identity factor *KLF4* (Table EV3). However, the majority of the upregulated mRNAs that we detected had not been described in association with T-cell stimulation (Table EV3). Of the 638 differentially expressed mRNAs, only ~20% were known to be involved in T-cell activation (Ellisen *et al*, 2001; Diehn *et al*, 2002; Cheadle *et al*, 2005). Among the newly detected upregulated genes were those that encode *GPR50, KLF4,*

*DUSP1, PPP1R15A, MASP2,* and *RGCC* proteins that are involved in processes such as MAPK signaling or other signaling pathways, the immune response, or the response to stimuli. Thus, the high sensitivity of TT-seq can uncover new target genes even in very well-studied systems.

### Defining the dynamic landscape of transcribed enhancers

The vast majority of transcripts with significantly changed synthesis after stimulation were ncRNAs. When we investigated the TT-seq coverage at known enhancers, we observed increasing RNA synthesis, showing that we could monitor eRNA production at transcribed enhancers such as the one at the *FOS* locus (Fig 2B). Within 15 min after stimulation, eRNA synthesis at this locus increased about 160-fold, whereas synthesis of *FOS* mRNA increased about 40-fold (Fig 2B). The TT-seq coverage profiles also immediately revealed bidirectional transcription at both the promoter and a known

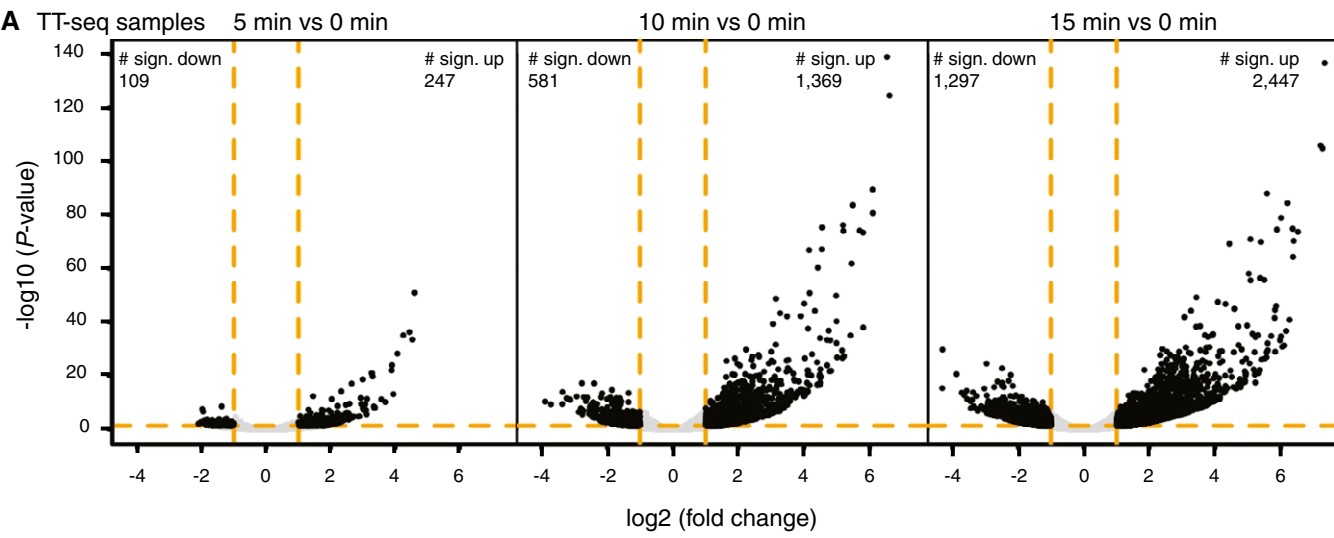

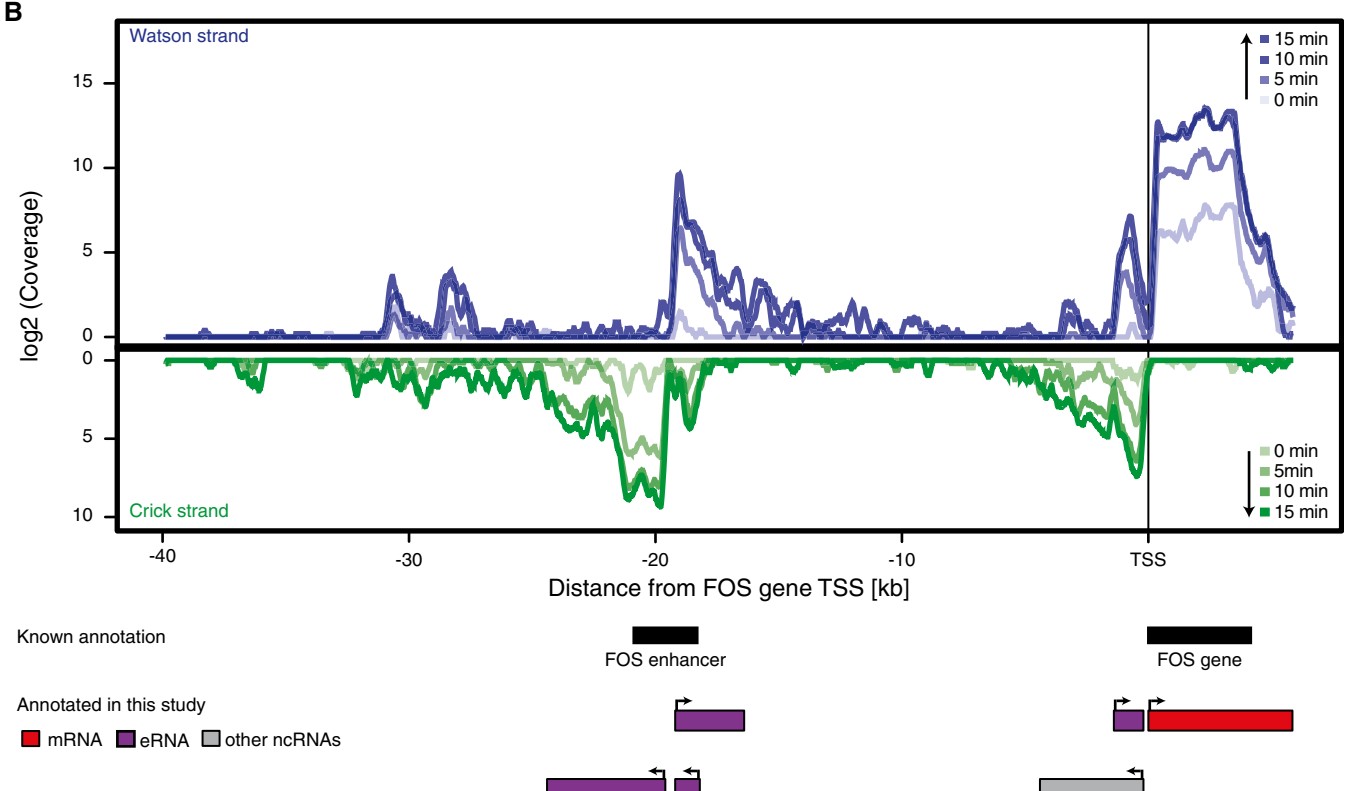

**Figure 2. TT-seq captures transcriptional changes after T-cell stimulation.**

A   TT-seq signal for statistically significant differentially expressed transcripts after 5, 10, and 15 min compared to time point 0 min, before cell stimulation. Significantly differentially expressed transcripts are indicated by black points. The numbers in the plots correspond to the numbers of significantly up/downregulated transcripts at each time point. The vertical orange lines indicate the fold change cutoff of 2, and the horizontal orange line shows the *P*-value cutoff of 0.05. *P*-values were derived via DESeq2 (Wald-Test).

B   Sense and antisense transcription at the *FOS* gene and its annotated upstream enhancer. The rectangles with arrows indicate transcripts annotated in this study.

enhancer at the *FOS* locus. Thus, enhancer transcription is very well captured by TT-seq, encouraging us to fully describe the landscape of transcribed enhancers and its changes during T-cell stimulation.

To select putative eRNAs from our annotated 12,673 ncRNAs, we used our recent GenoSTAN annotation of chromatin states in

genomes of 14 T-cell lines, which is based on the integration of publicly available chromatin marks and DNA accessibility data (Zacher *et al*, 2017). We compared all ncRNAs with all enhancer states from T cells (Zacher *et al*, 2017; Fig EV3). This resulted in 5,616 (44%) ncRNAs that overlapped with enhancer states either

with their transcribed region or with the region 1,000 bp upstream, and were therefore classified as putative eRNAs (Fig 3A).

**Immediate, nucleosome-depleted enhancers**

Out of a total of 50,810 annotated T-cell enhancer states, 7,865 produced eRNAs in our cell line and under our conditions that we could detect. The obtained putative 5,616 eRNAs showed a similar length distribution as the remaining 7,057 ncRNAs (Fig 3B and Appendix Fig S3A), but had shorter half-lives (Fig 3C and

Appendix Fig S3B), reflecting the known unstable nature of eRNAs. The sets of putative active eRNAs (applying the same cutoff as for the transcriptome annotation, RPK ≥ 16.5) comprises more than 5,000 actively transcribed eRNAs at each time point (Fig 3A). For a large fraction of eRNAs (29%), we observed significant changes in their synthesis during the time course compared to the initial time point (Materials and Methods), showing that eRNA transcription is highly regulated.

Consistent with the chromatin state annotation of enhancers, the putative eRNAs were flanked by a region of high DNase

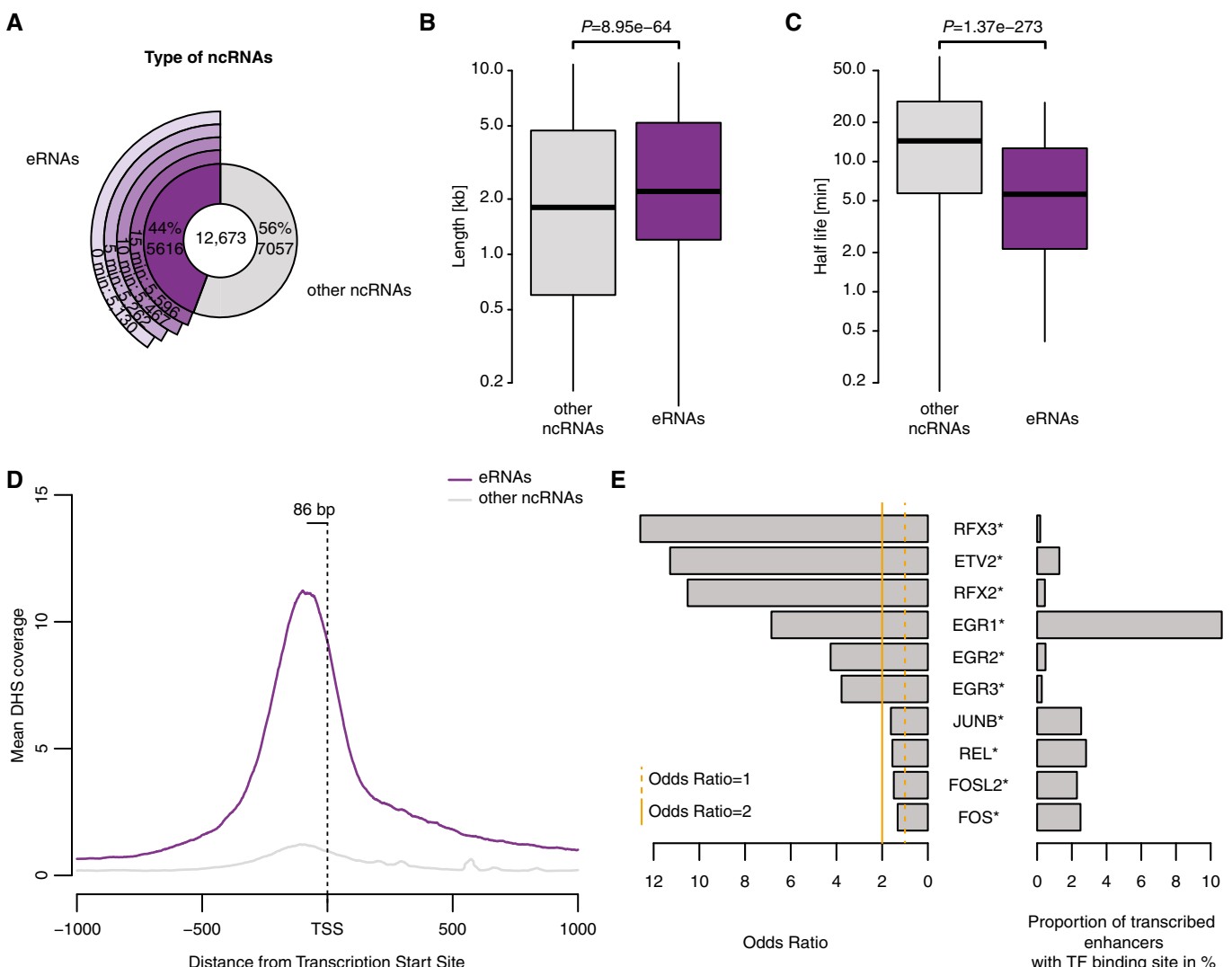

**Figure 3.  Characteristics of transcribed enhancers.**

A   Distribution of identified eRNAs among ncRNAs annotated based on TT-seq signal. The outer circle segments show the number of actively transcribed eRNAs (RPK ≥ 16.5) at each time point.

B   Length distribution of eRNAs and other ncRNAs. The *P*-value was derived by two-sided Mann–Whitney *U*-test. Box limits are the first and third quartiles, the band inside the box is the median. The ends of the whiskers extend the box by 1.5 times the interquartile range.

C   Half-life distribution of eRNAs and other ncRNAs. The *P*-value was derived by two-sided Mann–Whitney *U*-test. Box limits are the first and third quartiles, the band inside the box is the median. The ends of the whiskers extend the box by 1.5 times the interquartile range.

D   Average DNase hypersensitivity sites (DHS) signal at the TSS of eRNAs and other ncRNAs.

E   Motif enrichment in the 250 bp upstream sequences of eRNA versus other ncRNAs. Displayed are only motifs of upregulated TFs with odds ratio > 1.2 and *P*-value < 0.05. The stars indicate TFs with statistical significant enrichment upon eRNAs after multiple testing correction (Benjamini–Hochberg method, FDR < 0.05).

hypersensitivity (Encode Project Consortium, 2012) immediately upstream, and this was not the case for the remaining ncRNAs (Fig 3D). In addition, the region 250 bp upstream of the eRNA transcription start site (TSS) was significantly enriched for binding sites of transcription factors that act during T-cell activation, namely *EGR1, EGR2, ERG3, JUNB, REL, FOSL2, FOS* (odds ratios 6.8, 4.3, 3.8, 1.6, 1.6, 1.5, and 1.3, respectively), compared to other ncRNA upstream sequences (Fig 3E). This strongly indicates that our set of putative eRNAs represents transcripts originating from enhancers that are relevant for the T-cell response. Taken together, TT-seq can define the landscape of actively transcribed enhancers and its changes during T-cell stimulation.

### Transcription from promoters and enhancers is correlated and distance-dependent

We next paired eRNAs and mRNAs that localized within insulated neighborhoods (Fig 4A and Table EV4) that were defined with a combination of cohesin-ChIA-PET and CTCF-ChIP-seq profiling performed in the Jurkat cell line (Hnisz *et al*, 2016b). After removing pairs of upstream divergent (1 kb upstream of sense TSS) and convergent (1 kb downstream of sense TSS) transcripts and their bidirectional promoters, we obtained a total of 6,896 eRNA-mRNA pairs that represent putative enhancer–promoter pairs. These pairs contained 2,454 transcribed enhancers and 2,520 promoters. On average, there were 1.3 transcribed enhancers and 1.5 promoters located within an insulated region (Fig EV4A). The median transcribed enhancer–promoter distance within these pairs was 117 kb, with 52% of all paired transcribed enhancers residing within ± 50 kb from their closest paired promoter (Fig 4B). There was no preference for eRNA orientation with respect to the mRNA orientation (Fig EV4B). Due to the small size of the insulated neighborhoods and our conservative pairing, most transcribed enhancers (56%) and most promoters (72%) remained unpaired. The paired transcribed enhancers engaged on average with 2.8 promoters, whereas paired promoters engaged on average with 2.7 enhancers (Fig 4C).

We found that changes in RNA synthesis over time correlated very well between transcribed enhancers and their paired promoters (Fig 4D, Materials and Methods). When we shuffled the transcribed enhancers and promoters and paired them randomly, irrespective of insulated neighborhoods, the correlation dropped (Fig EV4C, *P*-value = 9.99e-4). Moreover, the correlation was higher for transcribed enhancers located < 10 kb from their paired promoter ("proximal enhancers") than for those located further apart ("distal enhancers") (Fig 4D). This indicates that enhancer transcription decreases with increasing distance from the activated target promoter, consistent with the observation that interacting enhancers tend to be close to their promoters (Dekker *et al*, 2013; He *et al*, 2014).

The distance between transcribed enhancer–promoter pairs is limited by the size of the insulated neighborhoods, but it is generally much shorter (*P*-value < 2.2e-16, Fig EV4D and E). Pairing within insulated neighborhoods leads to higher correlations than pairing every promoter with its closest transcribed enhancer (Appendix Fig S4). There is no relationship between the distance and the correlation over time between closest transcribed enhancer–promoter pairs (Spearman correlation −0.03,

Appendix Fig S5). When we splitted up the closest pairs dependent on their location within the same insulated neighborhood, the pairs within the same loop showed a higher correlation (*P*-value 0.00121, Appendix Fig S6). These results indicate that the correlation in changes of RNA synthesis from transcribed enhancers and promoters depends on both genomic distance and location within insulated neighborhoods.

### Rapid up- and downregulation via promoter–proximal elements

Our results raise the question how transcription can be activated from those promoters that do not have paired enhancers. It is known that promoters may contain proximal binding sites for transcriptional activators such as AP-1, a heterodimer of *FOS* and *JUN* proteins that is induced upon T-cell stimulation. Indeed, we found that upregulated but unpaired promoters were enriched for AP-1 binding sites (TGACTCA) in the promoter–proximal region 500- to 100 bp upstream of the TSS for mRNA transcription, compared to upregulated and paired mRNAs (odds ratio 2.24, *P*-value 0.031, Materials and Methods). This shows that TT-seq can be used to disentangle promoter-based from enhancer-based activation of gene expression.

TT-seq also revealed a large number of downregulated genes upon T-cell stimulation, as captured by ceasing RNA synthesis. Such rapid downregulation cannot be observed by RNA-seq, due to the stability of most mRNAs. When we compared sequence motifs for unpaired downregulated mRNAs compared to unpaired upregulated mRNAs, we did not find enriched motifs that are known to bind transcriptional repressors, but consistently found them to be depleted of binding sites for the transcriptional activator AP-1 in the region 500 to 100 bp upstream of the TSS (odds ratio 0.51, *P*-value 0.022, Materials and Methods). Together these observations show that TT-seq is ideally suited to detect downregulated genes and are consistent with the view that rapid gene regulation can be mediated by the promoter–proximal region.

### Transcription from enhancers and promoters occurs simultaneously

We next investigated whether there are temporal differences in the onset of transcriptional changes between enhancers and promoters. In particular, we wished to find out whether enhancers were transcribed before their paired promoters. To this end, we selected pairs where the transcriptional change 15 min after stimulation for both the transcribed enhancer and the promoter in the TT-seq samples was at least twofold up ("upregulated pairs") or twofold down ("downregulated pairs") and significant (FDR < 0.05). This selection ensures that both the promoters and the transcribed enhancer have been activated during the time course allowing to probe the relative timing of activation. The TT-seq data clearly showed that changes in RNA synthesis occurred simultaneously at paired transcribed enhancers and promoters, for both up- and downregulated pairs, at the temporal resolution of our data and within a given variation (Fig 5). This shows that for an immediate transcription response, the changes in RNA synthesis for enhancers and their paired promoters occur simultaneously, provided our current temporal resolution (Fig 5A and C).

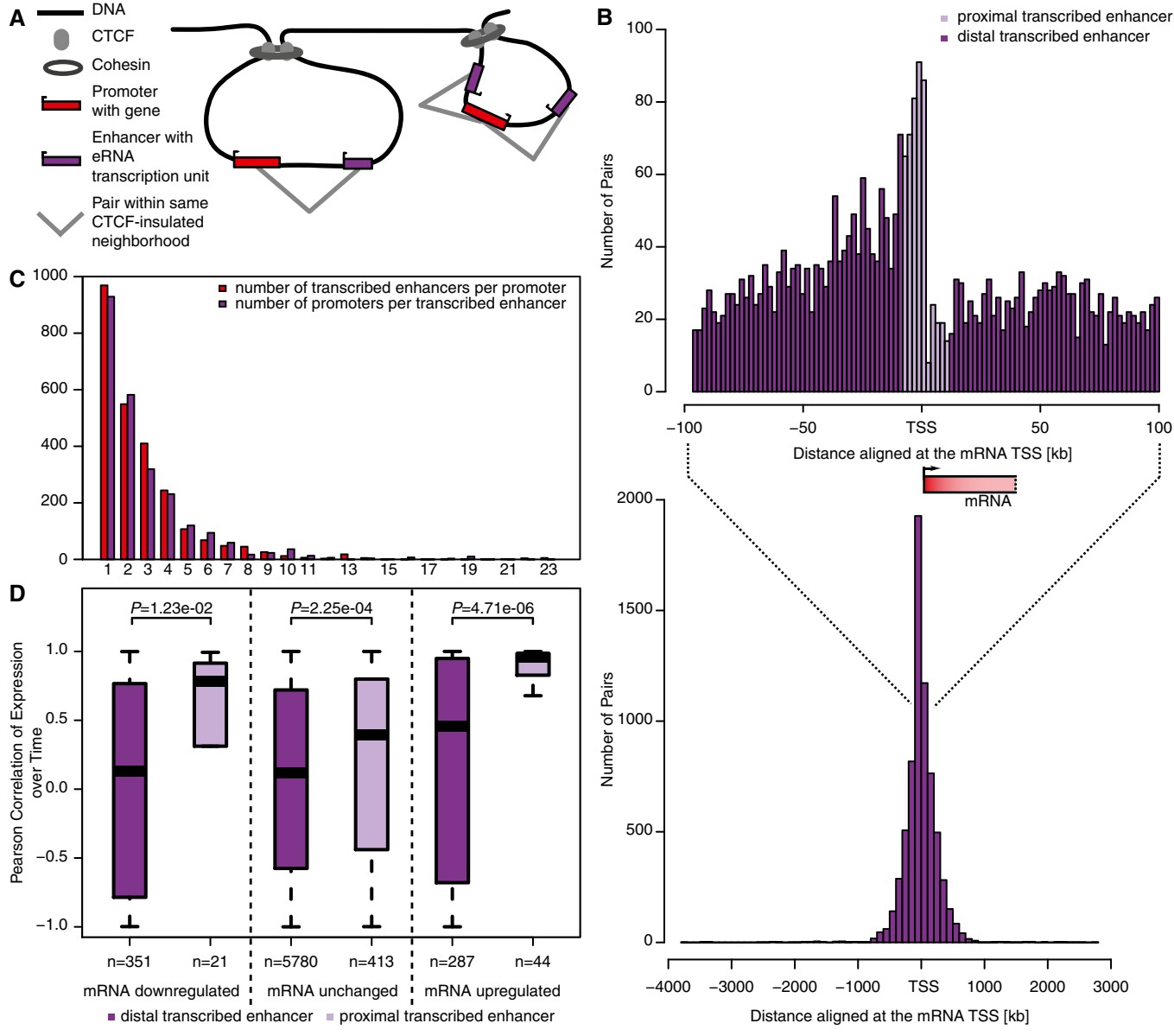

**Figure 4.   Pairings of transcribed enhancers with promoters.**

A   Schematic of transcribed enhancer–promoter pairing based on CTCF-insulated neighborhoods.

B   Distance distribution between eRNA and mRNA TSS. The lower histogram depicts the full distance range in 100-kb steps. The upper histogram shows a zoom-in of the region [TSS − 100 kb, TSS + 100 kb] in 2-kb steps. The position of the paired mRNA is indicated together with its median length.

C   Number of transcribed enhancers per paired promoter and promoters per transcribed enhancer.

D   Correlation of TT-seq signal over time between proximal (left, dark violet) or distal (right, light violet) transcribed enhancers and promoters by change in promoter TT-seq signal (from left to right: downregulated, unchanged, upregulated promoters). The Pearson correlation coefficient was calculated between read counts across the time series (replicates averaged per time point) for each transcribed enhancer–promoter pair. The *P*-values were derived by two-sided Mann–Whitney *U*-tests. Box limits are the first and third quartiles, the band inside the box is the median. The ends of the whiskers extend the box by 1.5 times the interquartile range.

In contrast, our RNA-seq data suggested that an increase in enhancer transcription preceded mRNA transcription for upregulated pairs (Fig 5B and D). However, this does not mean that eRNA synthesis changes more rapidly than mRNA synthesis. Instead, the half-life of eRNAs is around two orders of magnitude shorter than that of mRNAs (Rabani *et al*, 2014; Schwalb *et al*, 2016), and this renders eRNA levels very sensitive to changes in their synthesis

(Fig EV2). Also, RNA-seq cannot detect changes in downregulated pairs because mRNAs have long half-lives in the range of hours (Rabani *et al*, 2014; Schwalb *et al*, 2016), and therefore, a rapid shutdown in RNA synthesis does not change mRNA levels when monitored within minutes. Taken together, TT-seq enables monitoring rapid changes in both eRNA and mRNA synthesis that cannot be detected by RNA-seq in an unbiased manner.

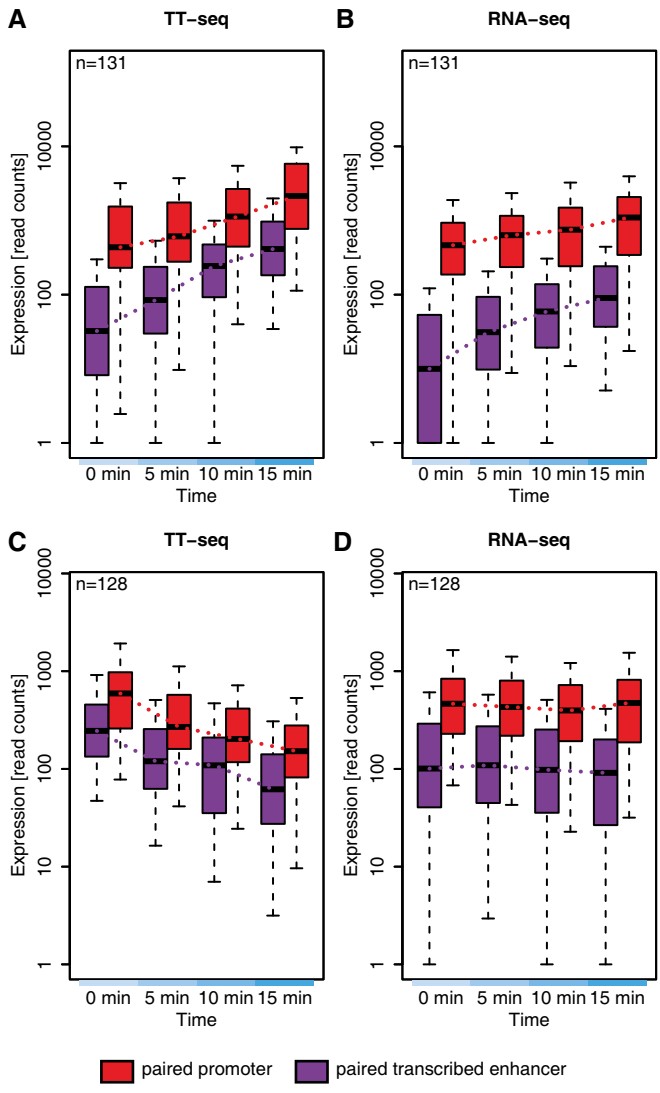

**Figure 5. Temporal changes in enhancer and promoter transcription.**

A  Development of TT-seq signal over time after T-cell stimulation for paired promoters and enhancers (*n* = 131) that are both significantly upregulated (FC ≥ 2, FDR ≤ 0.05) 15 min after stimulation (over the whole eRNA/the first 2,200 bp of the mRNA). The *y*-axis shows the normalized read counts over the whole transcribed enhancer region (violet) and the first 2,200 bp (average length of eRNA) of the paired mRNA (red). The black line indicates the median.

B  As in panel (A) but using RNA-seq read counts.

C  TT-seq signal change as in panel (A) but for paired promoters and enhancers (*n* = 128) that are both significantly downregulated (FC ≤ 1/2, FDR ≤ 0.05) 15 min after stimulation.

D  As in panel (C) but using RNA-seq read counts.

Data information: Box limits are the first and third quartiles, the band inside the box is the median. The ends of the whiskers extend the box by 1.5 times the interquartile range.

## Discussion

Here, we used TT-seq to monitor a very rapid transcription response in human T cells, and show that TT-seq can globally detect very short-lived transcripts such as eRNAs in a highly dynamic system at high temporal resolution. We demonstrate that TT-seq is suitable for annotating potential eRNAs and quantifying transcriptional changes very early after stimulation and thus provides insights into gene regulation, activation, and enhancer identity. Our results have implications for understanding the T-cell response, the temporal sequence of enhancer and promoter transcription during gene activation, the nature of functional enhancer–promoter pairing, and the design of future studies of transcription regulation in human cells.

First, our results provide new insights into the immediate T-cell response. TT-seq enabled us to detect immediate changes in the synthesis of thousands of transcripts. These RNAs included most of the transcripts known to be altered during T-cell stimulation, confirming known studies. We also found, however, many new mRNAs and ncRNAs that show altered synthesis upon T-cell stimulation. Many of these have functions in signaling pathways (*PPP1R15A, KLF4, ARC*), the response to stimuli (*KLF4, GPR50, DUSP1, MASP2*), or have catalytic activities (*DUSP1, MASP2*). Our results of immediate transcriptional changes confirm very early single-locus radio-labeled nuclear run-on studies of T-cell activation (Greenberg & Ziff, 1984). Our results thus extend and complement previous genome-wide studies of the T-cell response (Ellisen *et al*, 2001; Diehn *et al*, 2002; Cheadle *et al*, 2005) and help to more generally understand very early transcriptional responses.

Our work also identifies and characterizes eRNAs based on their synthesis, thereby mapping transcribed enhancers. We show that eRNA-producing enhancers can be paired with their target promoters by taking advantage of previously published data sets on insulated neighborhoods (Hnisz *et al*, 2016b) and chromatin states (Zacher *et al*, 2017). This yields enhancer–promoter pairs with highly correlated temporal changes in RNA synthesis. These results are consistent with the idea that eRNA transcription is a very sensitive and a good proxy for the activity of enhancers with respect to target gene activation (Hah *et al*, 2015). One limitation of the method, however, relates to the inability of TT-seq to detect intronic eRNAs in sense direction of mRNA transcription; however, only a small fraction of enhancers is missed this way.

The classification of non-coding RNAs remains challenging. The previous definitions of lincRNAs (long, stable, spliced, poly-adenylated) and eRNAs (short, short-lived, transcribed from enhancer element) do not always allow for a clear distinction between them (Paralkar *et al*, 2016). Here, we decided to exclude the GENCODE class of "lincRNAs" from our eRNA set because these are evolutionary conserved and less likely to be cell type-specific enhancer transcripts. Due to the low number of transcripts overlapping GENCODE-annotated lincRNAs (*n* = 590), we are not excluding many potential eRNAs, although some lincRNAs may stem from enhancers.

Our results also provide evidence that enhancer and promoter transcription can occur simultaneously during immediate gene activation. Our observations are derived from a single biological process with very fast response kinetics. Previous studies have observed that enhancer transcription precedes transcription from promoters, although in some cases, evidence for simultaneous transcription was also obtained (De Santa *et al*, 2010; Kaikkonen *et al*, 2013; Schaukowitch *et al*, 2014; Arner *et al*, 2015). These differences can to some extent be explained by the high sensitivity and temporal resolution of TT-seq, but may also reflect differences in the cellular responses monitored. Whereas we focused here on the immediate T-cell response that occurs within minutes, published work generally

analyzed responses after hours, and these require changes in chromatin at enhancers (Kaikkonen *et al*, 2013). Changes in chromatin could lead to a time lag between enhancer and promoter transcription and likely do not occur during the immediate response we investigated here because immediate-early genes responding within minutes are poised for gene activation, and chromatin is in a pre-open state (Tullai *et al*, 2007; Byun *et al*, 2009). Similarly, enhancers are primed for activity and are DNase I hypersensitive and modified with H3K4me1 (Wang *et al*, 2015).

Most importantly, our results demonstrate that TT-seq is an easy-to-use tool that is ideally suited to monitor rapid changes in the genomic landscape of transcribed enhancers and gene transcription in a non-perturbing manner *in vivo*. In addition to its high sensitivity and high temporal resolution, TT-seq is uniquely suited to detect immediate downregulation of genes, as it informs on drops in RNA synthesis when the mRNA product is long-lived and will give a signal in RNA-seq even at time points when transcription has been shut off for a long time already. In addition, TT-seq will map only those enhancers that produce eRNA at a certain time, providing apparently active enhancers rather than a list of all chromatin regions with enhancer signatures that may stem from past enhancer transcription events. TT-seq therefore facilitates the pairing of enhancers with putative target promoters. In the future, application of TT-seq to other human cells, signaling and differentiation events, is expected to provide novel biological insights into fundamental changes in gene regulatory programs.

# Materials and Methods

### TT-seq

Jurkat cells were acquired from DSMZ (Braunschweig, Germany). Cells were grown in RPMI 1640 medium (Gibco) supplemented with 10% heat-inactivated FBS (Gibco) and 1% penicillin/streptomycin (100×, PAA) at 37°C under 5% $CO_2$. Cells were labeled in media for 5 min with 500 μM 4-thiouridine (4sU, Sigma-Aldrich) and activated with 50 mM PMA (Sigma-Aldrich) and 1 μM ionomycin (Sigma-Aldrich). Cells were harvested, spike-ins were added, and RNA was purified and fragmented as described (Schwalb *et al*, 2016). Fragmented RNA was subjected to purification of labeled RNA as described (Dölken *et al*, 2008). Labeled fragmented RNA (TT-seq) and total fragmented RNA (Total RNA-seq) were treated with 2 units of DNase Turbo (Life Technologies). Sequencing libraries were prepared with the Ovation Human Blood RNA-seq library kit (NuGEN) following the manufacturer's instructions. All samples were sequenced on an Illumina HiSeq 1500 sequencer.

### Replicate measurements

We prepared TT-seq and total RNA-seq libraries for two biological replicates. For total RNA-seq, there were essentially no significant changes between time points, and the samples showed very high correlations and can be seen as replicates (see Appendix Fig S1). Replicate TT-seq libraries for time points 0 and 10 min after T-cell activation were obtained and showed high correlation (Spearman correlation coefficient 0.97, Appendix Fig S2). Based on these results, it was clear that the data are highly reproducible and of high

quality, making further replicate measurements obsolete. For all subsequent analyses, replicates were averaged after size factor normalization, where available. For transcriptome annotation, all TT-seq samples were used, irrespective of their sequencing depth, as GenoSTAN places more weight on deeper sequenced samples.

### Sequencing data processing

Paired-end 50-base reads with additional 6-base reads of barcodes were obtained for each of the samples. Reads were demultiplexed, and we could map 150–250 Mio read pairs per sample unambiguously with *STAR* (version 2.3.0) (Dobin & Gingeras, 2015) to the hg20/hg38 (GRCh38) genome assembly (Human Genome Reference Consortium) and the spike-in sequences. *Samtools* (Li *et al*, 2009) was used to quality filter SAM files, whereby alignments with MAPQ smaller than 7 (-q 7) were skipped and only proper pairs (-f99, -f147, -f83, -f163) were selected. Further data processing was carried out using the R/Bioconductor environment.

### Antisense correction

For merged transcribed regions from GENCODE, we selected strand-specific genomic regions where no antisense annotation existed in GENCODE. For all genomic positions in those regions, where the sense coverage exceeded 100 reads (i.e., highly expressed regions), we calculated the median ratio of antisense-to-sense coverage (including pseudo-count). This value provides an estimate of the antisense bias in every sample. We corrected the observed coverage/read counts for Watson and Crick strands, respectively, by solving the following formula, which assumes that the observed sense coverage is the sum of "real" sense coverage and a small percentage (i.e., the antisense bias value, called "$c$") of the "real" antisense coverage: $Coverage_{real}^{sense} = \frac{Coverage_{observed}^{sense} + c \times Coverage_{observed}^{antisense}}{1-c^2}$. For antisense correction of coverage profiles, the antisense coverage was averaged in a symmetrical 51-nt window around the position on the sense strand to normalize. For all further analyses (including calculation of expression values, fold changes, and synthesis/degradation rates), antisense-corrected feature counts (rounded to the nearest integer) were used.

### Transcription unit (TU) annotation and classification

Genomewide strand-specific coverage was calculated from fragment midpoints in consecutive 200-bp bins throughout the genome for all TT-seq samples. Binning reduced the number of uncovered positions within expressed transcripts and increased the sensitivity for detection of lowly synthesized transcripts. To overcome antisense bias due to highly expressed genes, an antisense correction was performed on each bin (as described in the previous paragraph). A pseudo-count was added to each bin to mask noisy signals. The R/Bioconductor package *GenoSTAN* (Zacher *et al*, 2017) was used to learn a two-state hidden Markov model with a PoissonLog-Normal emission distribution in order to segment the genome into "transcribed" and "untranscribed" states, which resulted in 139,507 transcribed regions. TUs that overlapped at least to 20% of their length with a protein-coding gene or a lincRNA annotated in GENCODE (gtf column "transcript_type" either "protein_coding" or "lincRNA") and overlapped with an exon of the corresponding annotated feature

were classified as protein-coding/lincRNA, and the rest was assumed to be ncRNAs. TUs mapping to exons of the same protein-coding gene/lincRNA were combined. In order to filter spurious predictions, a minimal expression threshold for TUs was defined based on overlap with genes annotated in GENCODE. The threshold was optimized using the Jaccard index criterion and resulted in 27,558 TUs with a minimal RPK of 16.5 (Fig EV1B). In order to overcome low expression or mappability issues, ncRNAs that are only 200 bp (1 bin) apart were merged. Subsequently, TU start and end sites were refined to nucleotide precision by finding borders of abrupt coverage increase or decrease between two consecutive segments in the two 200-bp bins located around the initially assigned start and stop sites via fitting a piecewise constant curve to the coverage profiles (whole fragments) for all TT-seq samples using the segmentation method from the R/Bioconductor package *tilingArray* (Huber *et al*, 2006). Overlapping transcripts (arising through overlaps with multiple annotated genes) were merged using the *reduce* function from the *GenomicRanges* package and assigned the corresponding protein-coding or lincRNAs id, if existing. 612 annotated transcripts (also used to calculate DESeq size factors) that overlapped with multiple protein-coding genes by at least 75% of the GENCODE transcript length and 20% of our transcript were removed from further analyses and are not reported in Table EV3, because they could not be clearly assigned to one gene. Protein-coding transcripts shorter than 5 kb and having overlap with less than 10% with any GENCODE protein-coding gene were classified as "ncRNA". All ncRNAs that started up to 1 kb downstream of a protein-coding gene on the sense strand were omitted in enhancer analysis or eRNA-ncRNA comparisons, as these reads might come from read-through transcription after the mRNA. This resulted in 22,141 non-ambiguously classified RNAs (8,878 protein-coding genes, 590 lincRNAs, and 12,673 ncRNAs, Table EV1), on which the rest of the analysis was focused. The class of eRNAs was comprised of 5,616 of our ncRNAs, where either the transcript or the region 1 kb upstream of the ncRNA overlapped with an enhancer annotated by GenoSTAN (Zacher *et al*, 2017) in at least one of the T-cell lines.

### Estimation of RNA synthesis rates and half-lives

To overcome inconsistent coverage throughout a gene due to splicing and multiple isoforms, constitutive exons (Bullard *et al*, 2010) were determined for all our mRNA and lincRNA transcripts. Read counts for those constitutive exons and all other ncRNA classes across all TT-seq and RNA-seq samples were calculated using *HTSeq* (Anders *et al*, 2014). To estimate rates of RNA transcription and degradation, we used the same approach as described in Schwalb *et al* (2016). Briefly, we used a statistical model that describes read counts $k_{ij}$ (in a TT-seq or RNA-seq sample) by the length of the feature (spike-in/transcript) $i$, $L_i$, and feature-specific labeled und unlabeled RNA amounts, $\alpha_i$ and $\beta_i$: $E(k_{ij}) = L_i\sigma_j(\alpha_{ij} + \epsilon_j\beta_{ij})$. We calculated the sequencing depths $\sigma_j$ and cross-contamination $\epsilon_j$ rates per sample $j$ based on the spike-in read counts, by setting $\alpha_{ij} = 1$ and $\beta_{ij} = 0$ for labeled spike-ins, and $\alpha_{ij} = 0$ and $\beta_{ij} = 1$ for unlabeled spike-ins. In a total RNA-seq sample, $\epsilon_j$ is fixed to 1, and in a TT-seq sample, $\epsilon_j$ is close to 0, as we enrich for labeled RNA. Then, this model was fitted by maximum likelihood to transcript read counts to provide estimates of the labeled and unlabeled RNA

amounts $\alpha_i$ and $\beta_i$ for a pair of TT-seq and RNA-seq measurements. The synthesis rate $\mu_i$ and the degradation rate $\lambda_i$ were calculated from $\alpha_i$ and $\beta_i$ assuming first-order kinetics as in Miller *et al* (2011) in the following way: $\lambda_i = -\frac{1}{t}\log\left(\frac{\beta_i}{\alpha_i+\beta_i}\right)$; $\mu_i = (a_i + \beta_i)\lambda_i$.

### Differential gene expression

Gene expression fold changes upon T-cell stimulation for each time point were calculated using the R/Bioconductor implementation of *DESeq2* (Love *et al*, 2014). The DESeq size factor was only estimated on our set of protein-coding genes. Differentially expressed genes were identified applying a fold change cutoff of 2 and an adjusted *P*-value cutoff of 0.05 comparing each time point to the 0 min measurements. For the absolute numbers of genes with changed synthesis, we checked if the TT-seq read count is significantly (adjusted *P*-value ≤ 0.05) changed at least twofold at any time point compared to time point 0 min.

### Motif analysis

DNA motifs in the form of PWMs were downloaded from the JASPAR database via the R/Bioconductor package *JASPAR2016* (Tan, 2015). Each PWM was screened against a positive and a negative set of sequences (e.g., 250 bp upstream sequences of eRNAs and remaining ncRNAs) with the *searchSeq* function in the *TFBSTools* package (Tan & Lenhard, 2016). We defined a cutoff to distinguish between motif occurrence and not-occurrence as 80% of the maximal score that the PWM could reach. The number of sequences in which the motif occurred was counted for the positive and negative set, and an odds ratio was calculated.

### eRNA-mRNA pairing

We paired all eRNAs and mRNAs in all possible combinations, as long as both transcript TSSs are within the same insulated neighborhood, defined by ChIA-PET Anchor sites (using the *findOverlaps* function from the *GenomicRanges* package; Lawrence *et al*, 2013). Pairs were removed, where the eRNA TSS fell into the region [TSS − 1,000; TSS + 1,000] around the protein-coding gene's TSS.

### External data processing

Experimentally validated enhancers were downloaded from the VISTA enhancer browser (http://enhancer.lbl.gov/frnt_page_n.shtml). ENCODE DNase-seq raw coverage files (for Fig 3D) and peak files (for Fig EV3) for Jurkat were retrieved from (https://genome.ucsc.edu/ENCODE/dataMatrix/encodeDataMatrixHuman.html) and replicates were merged. Enhancer and DHS coordinates were converted to hg20 coordinates using the *liftover* function in the R/Bioconductor package *rtracklayer* (Lawrence *et al*, 2009). ChIA-PET interaction domains processed with the Mango pipeline were downloaded from a previous study (Hnisz *et al*, 2016b) and were selected for *P*-values < 0.2. ChIA-PET Anchor sites were converted to hg20 coordinates using the *liftover* function in the R/Bioconductor package *rtracklayer* (Lawrence *et al*, 2009) followed by a *reduce* with *min.gapwidth = 60* which closes 90% of the gaps arising by *liftover*, in order to get continuous genomic regions.

## Data availability

The sequencing data sets have been deposited in the Gene Expression Omnibus (GEO) database under accession code GSE85201.

**Expanded View** for this article is available online.

## Acknowledgements

We thank Katja Frühauf and Michael Lidschreiber for stimulating discussions and Alex Graf for support with sequencing data pre-processing. CD was supported by a DFG fellowship through the Graduate School of Quantitative Biosciences Munich (QBM). JG was supported by the Bavarian Research Center for Molecular Biosystems and the Bundesministerium für Bildung und Forschung, Juniorverbund in der Systemmedizin "mitOmics" grant FKZ 01ZX1405A. PC was funded by Advanced Grant TRANSREGULON of the European Research Council (grant agreement No 693023), the Deutsche Forschungsgemeinschaft (SFB860, SPP1935), and the Volkswagen Foundation.

## Author contributions

MM carried out experiments. CD performed bioinformatics analyses. BZ provided scripts and assistance with transcriptome annotation. BS provided scripts and advice on data analysis. SK performed sequencing, supervised by HB. JG supervised bioinformatics. PC designed and supervised research. MM, CD, JG, and PC prepared the manuscript.

## Conflict of interest

The authors declare that they have no conflict of interest.

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
