## [Review Process File · Molecular Systems Biology]

TT-seq captures enhancer landscapes immediately after T-cell stimulation

Margaux Michel, Miss Carina Demel, Benedikt Zacher, Björn Schwalb, Stefan Krebs, Helmut Blum, Julien Gagneur and Patrick Cramer

*Corresponding authors: Julien Gagneur, Technische Universität München;
Patrick Cramer, Max Planck Institute for Biophysical Chemistry*

Review timeline:

Submission date:	16 December 2016
Editorial Decision:	23 January 2017
Revision received:	13 February 2017
Accepted:	15 February 2017

Editor: Maria Polychronidou

Transaction Report:

1st Editorial Decision

23 January 2017

Thank you again for submitting your work to Molecular Systems Biology. We have now heard back from the two referees who agreed to evaluate your study. As you will see below, the reviewers raise some concerns, which we would ask you to address in a revision.

The reviewers' recommendations are quite clear so I think that there is no need to repeat the points listed below. Please let me know in case you would like to further discuss any of the issues raised.

REFeree REPORTS

Reviewer #1:

This manuscript reports the application of TT-seq, a powerful method recently developed by the authors, to study rapid transcriptional responses of enhancers and promoters. It is demonstrated here that 4sU labeling pulses as short as 5 minutes can be applied to detect rapid up- and down-regulation of transcriptional activity genome-wide after application of particular stimuli to cells. Because TT-seq appears to be a relatively simple method (compared to some other run-on transcription mapping methods) these results are important to a broad readership and worthy of publication in MSB. The manuscript is clear and the data look solid.

Specific comments:

1. From the main text (incl Methods) it is unclear how many independent (?) biological replicates were done, and how these data were combined. How reproducible are the results between replicates? Scatterplots and correlation coefficients or similar metrics should be included (as Supp figures), so that readers get a clear sense of the reproducibility of the key results.
2. At some points the text is quite dry, as it is stuffed with counts of transcripts and other numbers, e.g. "TT-seq uncovered 247 up- and 109 down-regulated transcripts after 5 min, 1,369 up- and 581 down-regulated transcripts after 10 min, and 2,447 up- and 1,297 down-regulated transcripts after 15 min following stimulation", followed by "Out of a total of 3,744 transcripts that showed significantly changed synthesis 15 min after stimulation, 638 were mRNAs, and 2,986 were ncRNAs, including 120 lncRNAs. Of the mRNAs, 311 were up-regulated and 327 were down-regulated". Perhaps some of this detailed information can be summarized more globally and/or moved to figures or tables.
3. Please provide a reviewer link to the GEO accession, which is currently private.
4. p6. Please clarify how half-life of mRNAs was calculated based on these TT-seq data. It seems to me that a pulse-chase design is needed, but here only a pulse is given without the chase. How reliable/reproducible are these estimates? In Fig 3C half life is shown on an "arbitrary scale"; why not % per time unit?
5. Fig 3D. Y-axis reads "percentage ..." The maximum value is about 0.5 -- which would be very low as a percentage (about 1 in 200). Or should this be "fraction"? Furthermore, the text states "the putative eRNAs were flanked by a region of high DNase hypersensitivity" but ignores that ~50% (or 99.5%?) are not. A more balanced discussion is warranted.
6. I had hoped for a more critical attitude towards the simplistic assumption that "insulated neighborhoods" can be captured by taking CTCF-delineated ChIA-PET loops. In fact, it seems to me that the new TT-seq data offer an interesting opportunity to test how good this assumption is, compared to an alternative model in which simply distance between enhancer and promoter is the relevant parameter. Does the loop model indeed perform better in terms of correlations between enhancer and promoter activation? Such an analysis would be very informative for the "CTCF loop" community.

Reviewer #2:

The manuscript reports a very short time course stimulation (total 15 minutes) of Jurkat T cells followed with the TT-seq method developed by the authors of this manuscript. TT-seq seems to be suitable to broadly detect the immediate changes in the genome activity like promoters of mRNA and ncRNA/lncRNAs and enhancers. One of the main results is that at least a part of the immediately early enhancers and promoters are co-activated.

I have the following comments for the authors.

- 1) Due to the growingly importance of the lncRNA/ncRNAs, it is important to clarify what they are. The Gencode annotation has been a bit confused as well, but this requires clarification here. The authors report them as ncRNA and lncRNAs, reflecting previous annotations. However, are many of the ncRNAs essentially lncRNAs? And are many of the lncRNA/ncRNAs also enhancer RNAs, which may be stabilized? Again, while I understand that the Gencode does not help at first, I think that this issue should be clarified, at least by inspecting a number of lncRNA and ncRNAs. I would at least expect an in depth discussion about this.
- 2) In my understanding, there are 4 libraries produced without replication at each of the 4 time

points. The author can leverage of the similarity of the response at 5, 10 and 15 minutes (at least in Fig 1 and 2 in the shown examples we can see similar responses with different amplitude). However, the authors also suggest that eRNA transcription is highly variable (Page 8), however this needs either replication or validation of a number of them.

3) Importantly, the authors note that the activation of enhancers and promoters is simultaneous, and suggest that eRNAs are less stable. The authors suggest that conclusions are different from previous work, as cited in the manuscript, which suggested early activation of enhancer RNAs followed by promoter activation. We should be careful here for several reasons. (a) Actually, so short time intervals (5-15 minutes only activation) and Jurkat T cells activation are not used at least in Arner et al, one of the most comprehensive analysis so far (b) In the same Arner et al, there are several examples where there is "co-activation" of enhancers and promoters: the study contains many time points and biological, where many co-activation enhancer-promoter pairs can be seen and examples similar to this work may be seen. (c) Often, the mRNA expression peaks at much later stage (hours), in kinetics that are much longer.

1st Revision - authors' response

13 February 2017

Responses are in italics

Reviewer #1:

This manuscript reports the application of TT-seq, a powerful method recently developed by the authors, to study rapid transcriptional responses of enhancers and promoters. It is demonstrated here that 4sU labeling pulses as short as 5 minutes can be applied to detect rapid up- and down-regulation of transcriptional activity genome-wide after application of particular stimuli to cells. Because TT-seq appears to be a relatively simple method (compared to some other run-on transcription mapping methods) these results are important to a broad readership and worthy of publication in MSB. The manuscript is clear and the data look solid.

We would like to thank the reviewer for the support. We have gone through all of the points and updated the text and figures accordingly.

1. From the main text (incl Methods) it is unclear how many independent (?) biological replicates were done, and how these data were combined. How reproducible are the results between replicates? Scatterplots and correlation coefficients or similar metrics should be included (as Supp figures), so that readers get a clear sense of the reproducibility of the key results.

We are sorry that we had forgotten to add this information to the submitted manuscript. We have now included a paragraph in the Appendix methods and added scatterplots with correlations (Pearson correlation 0.97) as supplemental figures (Appendix Figures S1 and S2) for the time points where deep-sequenced replicates were collected and for total RNA at different time points (total RNA hardly changes within the first 15 minutes after stimulation). Generally, these analyses show that our data are highly reproducible and the quality of the data is very high.

2. At some points the text is quite dry, as it is stuffed with counts of transcripts and other numbers, e.g. "TT-seq uncovered 247 up- and 109 down-regulated transcripts after 5 min, 1,369 up- and 581 down-regulated transcripts after 10 min, and 2,447 up- and 1,297 down-regulated transcripts after 15 min following stimulation", followed by "Out of a total of 3,744 transcripts that showed significantly changed synthesis 15 min after stimulation, 638 were mRNAs, and 2,986 were ncRNAs, including 120 lncRNAs. Of the mRNAs, 311 were up-regulated and 327 were down-regulated". Perhaps some of this detailed information can be summarized more globally and/or moved to figures or tables.

We thank the reviewer for this comment and changed the text (p. 7) and figure (Fig 2) accordingly and added supplemental tables in the Appendix. We tried to move data from text to legends or

methods were possible, but some key numbers will have to remain in the main text.

3. Please provide a reviewer link to the GEO accession, which is currently private.

The GEO reviewer link is send by email to the editor, who we kindly ask to forward the link.

4. p6. Please clarify how half-life of mRNAs was calculated based on these TT-seq data. It seems to me that a pulse-chase design is needed, but here only a pulse is given without the chase. How reliable/reproducible are these estimates?

Indeed, it is not possible to estimate half-lives from TT-seq data alone. However, the combination of TT-seq data with total RNA-seq data enables us to estimate synthesis and degradation rates without a pulse-chase design. We have described this in detail elsewhere (Schwalb et al., Science 2016). Briefly, we use the information that total cellular RNA reflects the ratio of synthesis rate over degradation rate. As TT-seq gives an estimate for the synthesis rate, it is possible to determine the degradation rates from the RNA-seq data. We added an additional paragraph in the Appendix supplementary methods. This method was developed in the yeast system and there are multiple publications that use, test, and verify this approach. In particular, we had shown that the estimation of half-lives from 4sU-labeled RNA in combination with total RNA is less corrupted than rates derived by pulse-chase design due to shorter labeling times (Sun et al, Genome Research 2012).

In Fig 3C half life is shown on an "arbitrary scale"; why not % per time unit?

We corrected Figure 3C to show the half-life in absolute scale (minutes). We thank the reviewer for spotting this.

5. Fig 3D. Y-axis reads "percentage ..." The maximum value is about 0.5 -- which would be very low as a percentage (about 1 in 200). Or should this be "fraction"? Furthermore, the text states "the putative eRNAs were flanked by a region of high DNase hypersensitivity" but ignores that ~50% (or 99.5%?) are not. A more balanced discussion is warranted.

We see that this notion was misleading and thank the reviewer for pointing this out. Indeed we meant "fraction" and corrected accordingly. As the "presence of a region of high DNase hypersensitivity (=peak)" is based on a cutoff (for peak detection), we decided to change the figure in the revised manuscript and show raw coverage instead. We also carefully edited the text to reflect this.

6. I had hoped for a more critical attitude towards the simplistic assumption that "insulated neighborhoods" can be captured by taking CTCF-delineated ChIA-PET loops. In fact, it seems to me that the new TT-seq data offer an interesting opportunity to test how good this assumption is, compared to an alternative model in which simply distance between enhancer and promoter is the relevant parameter. Does the loop model indeed perform better in terms of correlations between enhancer and promoter activation? Such an analysis would be very informative for the "CTCF loop" community.

We thank the reviewer for this comment and agree that this is a partially open question. We think enhancer-promoter pairing within "CTCF loops" is less stringent than selecting the closest neighbor regarding genomic coordinates. Also, topological changes upon treatment could account for varying regulation of a promoter by different enhancers. To answer the question if pairing within CTCF loops improves correlation over pairing between close neighbors, we conducted additional analyses. We compared the correlation of enhancer-promoter pairs within "CTCF loops" and the correlation of synthesis changes for each mRNA and their closest eRNA. We see that the loop model leads to clear improvement of the correlation. Also closest enhancer-promoter pairs within a CTCF loop have a higher correlation than other closest pairs. We included supplemental figures in the Appendix and revised the manuscript (p. 9) accordingly.

Reviewer #2:

The manuscript reports a very short time course stimulation (total 15 minutes) of Jurkat T cells followed with the TT-seq method developed by the authors of this manuscript. TT-seq seems to be suitable to broadly detect the immediate changes in the genome activity like promoters of mRNA and ncRNA/lncRNAs and enhancers. One of the main results is that at least a part of the immediately early enhancers and promoters are co-activated.

We thank the reviewer for the careful evaluation and insightful comments.

1) Due to the growingly importance of the lncRNA/ncRNAs, it is important to clarify what they are. The Gencode annotation has been a bit confused as well, but this requires clarification here. The authors report them as ncRNA and lncRNAs, reflecting previous annotations. However, are many of the ncRNAs essentially lncRNAs? And are many of the lncRNA/ncRNAs also enhancer RNAs, which may be stabilized? Again, while I understand that the Gencode does not help at first, I think that this issue should be clarified, at least by inspecting a number of lncRNA and ncRNAs. I would at least expect an in depth discussion about this.

We are sorry if this was not clear. We called lincRNA all transcribed regions overlapping at least 20% with GENCODE-annotated lincRNAs. We called ncRNAs all the remaining transcribed regions non-overlapping coding genes. We next further classified the ncRNA as eRNAs when they overlapped an enhancer region. We understand the text needed clarification about this, which we did now (p. 6). We also checked that the labels in the supplemental figures are fine.

In addition, it is not easy to distinguish lincRNAs from eRNAs (Espinosa, Mol Cell 2016, and Paralkar, Mol Cell 2016). Hence, it is true that GENCODE-annotated lincRNAs could potentially also be eRNAs, and a large fraction (385/590) indeed overlaps with enhancer states in T cell lines. A conservative approach is to omit GENCODE-annotated lincRNAs from our eRNA analysis. We added a paragraph in the discussion (p. 12) and produced another Appendix Figure (S3), that shows difference between our lincRNA and eRNA classes, both in terms of length and half-life. We have expanded the discussion and this should address the concern.

2) In my understanding, there are 4 libraries produced without replication at each of the 4 time points. The author can leverage of the similarity of the response at 5, 10 and 15 minutes (at least in Fig 1 and 2 in the shown examples we can see similar responses with different amplitude). However, the authors also suggest that eRNA transcription is high variable (Page 8), however this needs either replication or validation of a number of them.

We are sorry that the information on replicates was missing. We included a paragraph and supplemental figures in the Appendix. Indeed, we show that our TT-seq data is highly reproducible. The statement on p. 8 that this reviewer refers to was "For a large fraction of eRNAs (29%) we observed significant changes in their synthesis between time points (Methods), showing that eRNA transcription is highly variable." This concerns changes with respect to time point 0 that are statistically significant and indicate that eRNAs are highly regulated. We rewrote the sentence to avoid such confusion in the future. Please also compare our answer to reviewer #1.

3) Importantly, the authors note that the activation of enhancers and promoters is simultaneous, and suggest that eRNAs are less stable. The authors suggest that conclusions are different from previous work, as cited in the manuscript, which suggested early activation of enhancers RNAs followed by promoter activation. We should be careful here for several reasons. (a) Actually, so short time intervals (5-15 minutes only activation) and Jurkat T cells activation are not used at least in Arner et al, one of the most comprehensive analysis so far (b) In the same Arner et al, there are several examples where there is "co-activation" of enhancers and promoters: the study contains many time points and biological, where many co-activation enhancer-promoters pairs can be seen and examples similar to this work may be seen. (c) Often, the mRNA expression peaks at much later stage (hours), in kinetics that are much longer.

We thank the reviewer for this comment. We have considerably revised and extended the text (p. 12) to account for the concerns of the reviewer. We agree that the main difference in these studies is the time frame; whereas we investigate immediate changes after activation within minutes, the published work generally investigates changes hours after stimulation, which allows for strong changes in the chromatin landscape. Note that when we cite the Arner paper we say that enhancer transcription CAN precede promoter transcription, not that it DOES, consistent with the co-activation pointed out by the reviewer. We made sure the text correctly reflects this. We do not wish to argue that there are any issues with the prior work and trust this message is correctly conveyed.

2nd Editorial Decision

15 February 2017

Thank you for sending us your revised manuscript. We think that the points raised by the referees have been satisfactorily addressed and I am pleased to inform you that your paper has been accepted for publication.

Corresponding Author Name: Patrick Cramer

Manuscript Number: MSB-16-7507